# Genetic Programming-Based Feature Construction for System Setting Recognition and Component-Level Prognostics

Francesca Calabrese *, Alberto Regattieri, Raffaele Piscitelli, Marco Bortolini and Francesco Gabriele Galizia

Department of Industrial Engineering (DIN), University of Bologna, 40136 Bologna, Italy; alberto.regattieri@unibo.it (A.R.); raffaele.piscitelli3@unibo.it (R.P.); marco.bortolini3@unibo.it (M.B.); francesco.galizia3@unibo.it (F.G.G.)
* Correspondence: francesca.calabrese9@unibo.it

**Abstract:** Extracting representative feature sets from raw signals is crucial in Prognostics and Health Management (PHM) for components' behavior understanding. The literature proposes various methods, including signal processing in the time, frequency, and time–frequency domains, feature selection, and unsupervised feature learning. An emerging task in data science is Feature Construction (FC), which has the advantages of both feature selection and feature learning. In particular, the constructed features address a specific objective function without requiring a label during the construction process. Genetic Programming (GP) is a powerful tool to perform FC in the PHM context, as it allows to obtain distinct feature sets depending on the analysis goal, i.e., diagnostics and prognostics. This paper adopts GP to extract system-level features for machinery setting recognition and component-level features for prognostics. Three distinct fitness functions are considered for the GP training, which requires a set of statistical time-domain features as input. The methodology is applied to vibration signals extracted from a test rig during run-to-failure tests under different settings. The performances of constructed features are evaluated through the classification accuracy and the Remaining Useful Life (RUL) prediction error. Results demonstrate that GP-based features classify known and novel machinery operating conditions better than feature selection and learning methods.

**Keywords:** prognostics and health management; feature construction; genetic programming

## 1. Introduction

Prognostics and Health Management (PHM) relies on collecting and analyzing signals produced by a system to determine its health status at any moment during its functioning and in the future. The two primary goals of PHM are diagnostics and prognostics. Diagnostics deals with detecting, isolating, and identifying a fault when it occurs [1]. Hence, it deals with defining the health condition of a given system or component at the moment of signal acquisition. Prognostics aims to predict the Remaining Useful Life (RUL) of the system or component [2] and, therefore, it deals with defining when the fault will occur.

In a data-driven approach, given the weak information about the system health condition provided by raw signals, it is necessary to process and transform raw signals before applying Machine Learning (ML) for diagnostics and prognostics [3]. In particular, besides noise removal, it is essential to extract relevant and non-redundant features from raw signals. A feature is relevant when it can point out the information about the system's health condition, usually hidden in the raw signal. A set of features is non-redundant when each feature in the set contributes in a distinct way to increase the informative content. A so-characterized feature set is beneficial to diagnostics and prognostics because it may facilitate the learning process of ML algorithms while increasing their generalizability and reducing their computational complexity. Relevant features are usually obtained through signal processing in the time, frequency, or time-frequency domain. The most common

approach consists of dividing the signal into segments of defined length and computing one or more features for each signal segment. Depending on the failure propagations, features may assume different trends, e.g., strictly monotonic, constant, mixed, or constant with discontinuities [4]. Thus, a deep knowledge of the system is required to identify relevant features representing specific failure propagations. Alternatively, dimensionality reduction methods, including feature selection and feature learning, can be implemented to extract relevant features from a large original set according to a specific object and, at the same time, to reduce the eventual redundancy [5,6].

Methods for feature selection extract relevant features from the original set without any transformation [7]. In particular, filter methods rank the features in the original set according to one or more evaluation criteria and select the features with the highest score [8,9]. Wrapper and embedding feature selection approaches use ML algorithms' performances to select the optimal subset of features [10,11]. On the other side, feature learning methods project the original feature set into a new feature space with a lower dimension, losing the physical meaning of the original features [12]. Deep Learning (DL), which allows extracting a relevant and non-redundant feature set automatically and in an unsupervised way, represents a breakthrough in feature learning methods. DL allows the construction of end-to-end diagnostics and prognostics models to learn features directly from raw data and recognize and predict the health state of machines [13,14].

The major problem with DL-based feature learning is that it requires many calculations during the training process and many hyperparameters to set [15].

An emerging task in data science is Feature Construction (FC), which aims to construct higher-level features from the original feature set to increase the classification performance and eventually reduce the dimensionality of features [16]. Like feature learning, FC may provide a combination of the original features in output and, therefore, the resulting features may not have a clear physical meaning. However, FC may also work just as feature selection when the original feature set contains a feature that already describes the failure propagation. In addition, FC can be used for both diagnostics and prognostics since it allows constructing features according to specific criteria to satisfy, e.g., similarity and monotonicity [17].

Genetic Programming (GP) represents an effective tool for FC. It automatically selects and constructs more discriminating features without requiring a pre-defined model and a high amount of training data [18]. Feature selection and construction are performed according to a fitness function, which can be designed depending on the objective of the analysis, e.g., classification, clustering, and regression.

This paper uses GP to construct feature sets for two distinct goals, i.e., system setting detection and prognostics. System setting detection aims to recognize the specific operating condition under which the system is working. An optimal prognostics model, and therefore an optimal HI, should be insensitive to the implemented setting. However, industrial data are often unlabelled and not provided with contextual information, especially if the perspective of a machine producer is assumed [19]. These characteristics make it necessary to detect a change in the system setting to understand, for instance, that a different trend in the extracted feature does not correspond to a component fault. Like diagnostics, system setting detection is a pattern recognition problem and, therefore, can be faced through classification and clustering ML algorithms [20]. Since not all machinery settings may be known at the time of the analysis, two distinct GPs are built for the setting recognition goal in this paper. The first GP uses a supervised algorithm, i.e., the k-Nearest Neighbor (k-NN), to evaluate the feature constructed at each iteration; therefore, the label indicating the system setting is used during the assessment process. The second GP uses the k-Means algorithm, which adopts an unsupervised learning approach and does not require any information on the implemented setting. Both GPs provide one only feature, which should be as constant as possible within the same class/cluster since not influenced by the single component behaviors. The performances of constructed features are evaluated in terms of the classification accuracy obtained by two ML supervised algorithms, i.e., the Decision

Tree (DT) and the k-NN. Finally, a third GP is built for the prognostics goal. It provides a component-level Health Indicator (HI) that describes the components' degradation process. The HI should have a clear trend and similar values at the end of the components' life, i.e., similar Failure Thresholds (FT). To this aim, a multi-criteria fitness function considering monotonicity, trendability, and prognosability is used to assess the constructed HI.

The proposed GP models are applied to vibration signals collected from a test rig built in the Department of Industrial Engineering of the University of Bologna.

The main contributions of this paper can be summarized as follows. First, FC is performed to build two distinct features, one identifying the system setting and the other one describing the failure progression at a component level. Hence, a simultaneous analysis at a system and component level is conducted. Second, to the best of our knowledge, it is the first time that a clustering-based fitness function for system setting recognition is included in the GP in the prognostics field.

The remaining of this paper is organized as follows. Section 2 reviews the existing literature about GP-based FC for diagnostics and prognostics. In Section 3, the theoretical background and critical elements of GP are described. In addition, the adopted fitness functions for setting detection and prognostics are described. Finally, a case study shows the application of the developed GPs to vibration signals collected during several run-to-failure tests.

## 2. Materials and Methods

Many AI optimization algorithms for searching a function maximum (or minimum) value work in a finite domain, consider multiple constraints on the solution set, and have issues if the objective function has multiple local maxima or non-linearity trends [21]. In particular, these algorithms require an unacceptable amount of time to reach the optimal solution. Hence, the attention moves towards heuristic algorithms, which can guarantee sub-optimum solutions to the problem in a reasonable time. Effective heuristic algorithms are Evolutionary Algorithms (EA), which have their fundamentals in Darwin's evolutionary theory, particularly Genetic Programming (GP).

GP emulates the evolution of a population's individuals through genetic operators such as crossover and mutation [22]. The individuals represent the possible solutions belonging to the population, and their strength is evaluated by a fitness function, which expresses their ability to adapt to the environment. Thus, only individuals with high fitness function values survive during the construction process. The main steps of GP can be summarized as follows [23]:

1. First, an initial population of individuals is randomly generated
2. Then, the following steps are performed until a specific termination criterion is met
   a. A fitness value is assigned to each individual
   b. The individuals with the best fitness value are selected and reproduced for the next generation
   c. A new population is created through genetic operators
   d. The result of genetic operators represents a possible solution to the generation

Typical termination criteria are the fitness function threshold and the number of generations. Genetic operators for individuals' evolution are reproduction, crossover, and mutation. The reproduction operator consists of copying an individual (chosen for his fitness score) into the new population without any transformation. The crossover operator introduces variation in the population by creating offspring that includes some parts of their parents, chosen by a selection method. The mutation operator is a stochastic alteration of one or more genes. It introduces the exploration of new spaces on the fitness surface to avoid the premature convergence of the program. Each operator is realized with a certain probability. In particular, the probability of the mutation operator $P_m$ cannot be greater than 0.1. Between the operators' probabilities, the following relation is held:

$$P_r + P_c + P_m = 1 \tag{1}$$

where $P_r$, $P_c$, $P_m$ represent the probability of reproduction, crossover, and mutation, respectively.

At each iteration, the GP creates a program that can have a tree-based representation. The nodes represent the function set, which contains all the operator types, e.g., mathematical, arithmetic, Boolean, conditionals, and looping. Typical elements of the function set are summarized in Table 1 [24]. The tree's leaves represent the terminal set, which includes the variables to combine with the operators and sometimes constant values.

**Table 1.** Function set.

| Kind of Primitives | Example(s) |
| --- | --- |
| Arithmetic | Add, Multiplication, Division |
| Mathematical | Sin, Cos, Exp |
| Boolean | AND, OR, NOT |
| Conditional | IF-THEN-ELSE |
| Looping | FOR, REPEAT |

Other essential parameters of GP are the generation gap, which is defined as the percentage of the population that survives from one generation to another, and the parents' selection methods, often chosen between tournament selection and roulette-wheel selection [25].

Genetic Programming for feature construction is receiving significant attention because of its flexibility and adaptability in different contexts [26].

According to data scientists, a feature set constructed through GP may significantly increase classification or clustering performances depending on the formulated fitness function. In the case of classification, the fitness function mainly includes genetic algorithms [27]. In the case of clustering, the fitness function can include different measures, such as the connectedness [28] or the silhouette coefficient [29]. A clustering algorithm is embedded into the GP algorithm to get data partitions in both cases.

Although the promising results for the classification and clustering tasks in the data science domain, few works use GP-based feature construction for diagnostics. Firpi et al. [30] use GP to construct artificial features for fault detection using the Fisher rate as the fitness metric. Peng et al. [31] propose a fault diagnosis approach based on the extraction, construction, and combination of features through GP that uses classification results of the k-NN algorithm in the fitness function. On the contrary, feature construction through GP is mainly investigated for prognostics to identify degradation indicators that reveal the failure progression [32]. Unlike features for diagnostics, which have to be as much different as possible for distinct health conditions, an optimal HI should have peculiar characteristics. Nguyen et al. [33] provide an overview of the HI evaluation criteria for prognostics, divided into two categories: the first looks at the performance of the HI construction phase, and the second focuses on prognostic results. Many existing works use the evaluation criteria of the first category and, in particular, the intrinsic characteristic of the HI, such as monotonicity, trendability, and failure consistency. Liao [34] proposes a GP-based HI construction approach for automatically identifying prognostics features for RUL prediction. They extract statistical features in the time domain first. Then a single-objective fitness function, including monotonicity, is introduced to construct an optimal HI. Similarly, Qin et al. [32] adopt monotonicity as the fitness function for prognostics of rotating machinery. Wang et al. [24] use a GP-based HI construction for rolling bearings prognostics. In this case, the original feature set includes features in the time, frequency, and time–frequency domains, and the fitness function is equal to the arithmetic mean of three terms, i.e., monotonicity, trendability, and deviation. Nguyen et al. [33] propose a multi-criteria GP-based HI construction methodology, in which the input feature set consists of 11 time-domain features, and the fitness function is represented by a combination of eight different HI evaluation criteria. Wen et al. [35] include GP in the stochastic process for RUL prediction. In particular, considering the characteristics of the degradation models,

two properties of the HI are identified, i.e., consistency and the average range of HI, and considered in the fitness function.

Similar to the works mentioned above, this paper uses a GP for HI construction based on three metrics, i.e., monotonicity, trendability, and prognosability. In addition, GP is also used for constructing system-level features and classifying the system setting.

Table 2 summarizes the main approaches adopted in the related literature for feature extraction, feature construction, and HI construction through Genetic Programming. The first difference of the present study with respect to related works lies in the use of GP for two distinct purposes, i.e., features construction and HI construction, which allow realizing a simultaneous analysis at a system level and a component level. Indeed, features with different trends are built from the same feature set consisting of time-domain features extracted from raw signals depending on the fitness function included in the GP training. In particular, a classification- or cluster-based fitness function addresses the machinery setting recognition issue, while a fitness function based on typical metrics for RUL prediction address the component-level prognostics issue. In addition, because the working conditions are not always known, both classification- and clustering-based fitness functions for the setting recognition. In this way, GP can also be used in case of scarce availability of labeled datasets, which is a widespread situation for industrial machinery producers.

**Table 2.** Similarities and differences between the proposed approach and other works.

| References | Feature Extraction | Fitness Function | |
| --- | --- | --- | --- |
| | | Features Construction | HI Construction |
| [30] | - | Fisher ratio | - |
| [31] | Embedded in the function set | k-NN | - |
| [32] | 7 Time-domain features | - | Monotonicity |
| [33] | 11 Time-domain features | - | Monotonicity, trendability, failure consistency, scale similarity, robustness, mutual information Spearman correlation with RUL, F-test |
| [34] | 68 Time- and frequency-domain features | - | Monotonicity |
| [35] | - | | Weighted sum of failure consistency and the average range |
| [24] | 21 Time, frequency, and time–frequency domain features | - | Arithmetic mean of monotonicity, trendability, and deviation quantity |
| This paper | 81 Time-domain features | k-NN Silhouette index | Weighted mean of monotonicity, trendability, and prognosability |

### 2.1. Test Rig Description and Data Collection

This section describes the methodology followed to construct system-level features for system setting recognition and component-level features for prognostics.

The system under analysis, which is shown in Figure 1, is a test rig built in the Department of Industrial Engineering of the University of Bologna. It consists of an asynchronous motor, a gearbox made of two pulleys exchanging the rotational motion through a belt, two shafts that share the torques due to a couple of gears, and an electromagnetic brake. The three-phase electric motor is an eight poles motor with 0.23 kW power and possible rotation speeds equal to 710 rpm and 910 rpm. The motion is transmitted to the first shaft through the belt running on the pulleys and put in tension thanks to a screw-system positioned on the motor's support, allowing a regulation of the belt's tension which is essential for accelerated run-to-failure tests. The braking system consists of an electromagnetic dust brake with adjustable braking torque in 0–7.5 Nm and a 90 VDC command control fed by a transformer. The platform's data acquisition device is composed of three accelerometers and a pyrometer. Three Dytran 3093D3 triaxial accelerometers are placed on the bearing's support, next to the second pulley and the two gearboxes. The three sensors have

a sampling frequency of 12.8 kHz per axis and an acceleration range of 500 Gpeak. An OPTRIS CSmicro infrared sensor is placed near the second pulley measuring the pulley or the belt's temperature at a sampling frequency of 1 kHz. The accelerometers are connected to a computer through three-channel NI9230 I/O modules with a maximum sampling rate of 12.8 kS/s for each channel mounted on a four-slot NI 9274 chassis that collects all the data from the acceleration sensors before sending it to the computer through a USB connection. A data acquisition interface is placed in the pyrometer's cable, and temperature measurements are collected using the plug-n-play software CompactConnect supplied by the Optris company. The technical characteristics of all components included in the test rig can be found in Appendix A.

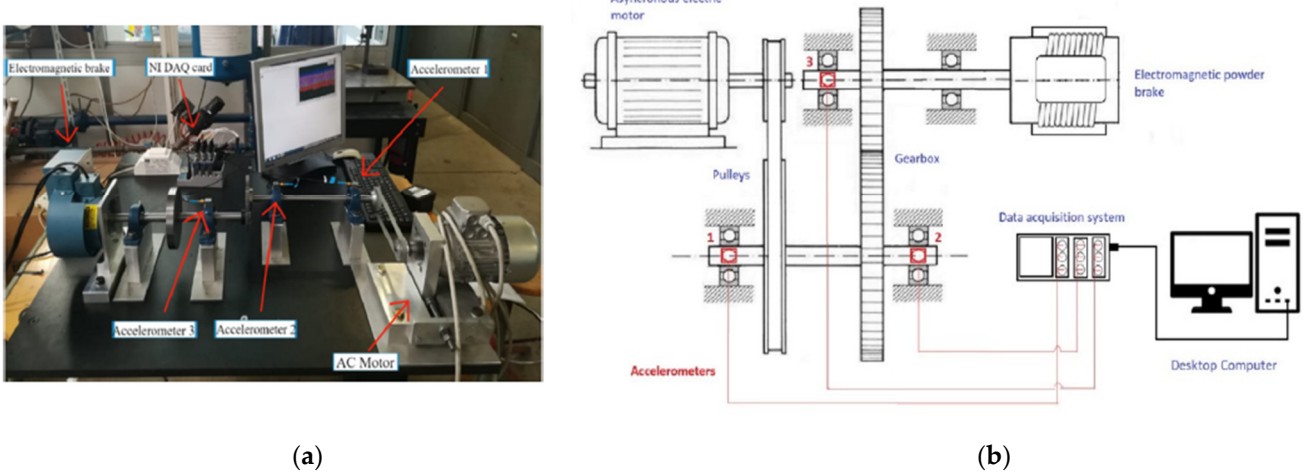

(**a**)                                                                                 (**b**)

**Figure 1.** (**a**) The test rig and (**b**) its mechanical scheme.

The component under analysis for prognostics is the belt. In particular, six run-to-failure tests are conducted, in which the belt is stressed until its failure. The more significant stress is caused by putting the belt at its maximum stretch to accelerate its degradation until it suddenly breaks. A greater tension of the belt usually occurs because of an improper mounting, which causes faster wear of the component and consequently more system downtimes. Tests are conducted in three different operating conditions, named C1, C2, and C3, defined by distinct braking torque values and the motor speed (Table 3). The braking force values have been set to 0.1 and 0.2 Nm because of the low power of the electric motor. Indeed, higher values would cause an overheating of the motor until its failure. In addition, a major braking force would cause fewer vibrations in the system, which would slow down the degradation process and make tests much longer. The motor speed values have been chosen because of technical aspects. For each setting, one or two belt run-to-failure tests are conducted. In addition, a run-to-failure test is also conducted with varying operating conditions. Run-to-failure tests for each condition are described in Table 4.

**Table 3.** Setting parameters.

| Operating Condition | AC Motor Speed (rpm) | Braking Force (Nm) |
|:---:|:---:|:---:|
| C1 | 710 | 0.1 |
| C2 | 710 | 0.2 |
| C3 | 910 | 0.2 |

**Table 4.** Available datasets.

| Test | Setting | Run-to-Failure Trajectories | Duration (min) |
|------|---------|-----------------------------|----------------|
| 1 | C1 | F1 | 38.4 |
| 2 | C1 | F2 | 170.5 |
| 3 | C2 | F3 | 157.9 |
| 4 | C2 | F4 | 74.2 |
| 5 | C1-C2 | F5 | 328.1 + 667.1 |
| 6 | C1-C2-C3 | F6 | 208.9 + 232.2 + 995.5 |

### 2.2. The Methodology

The methodology consists of two main steps: the training phase, whose framework is depicted in Figure 2, and the testing phase, which is depicted in Figure 3.

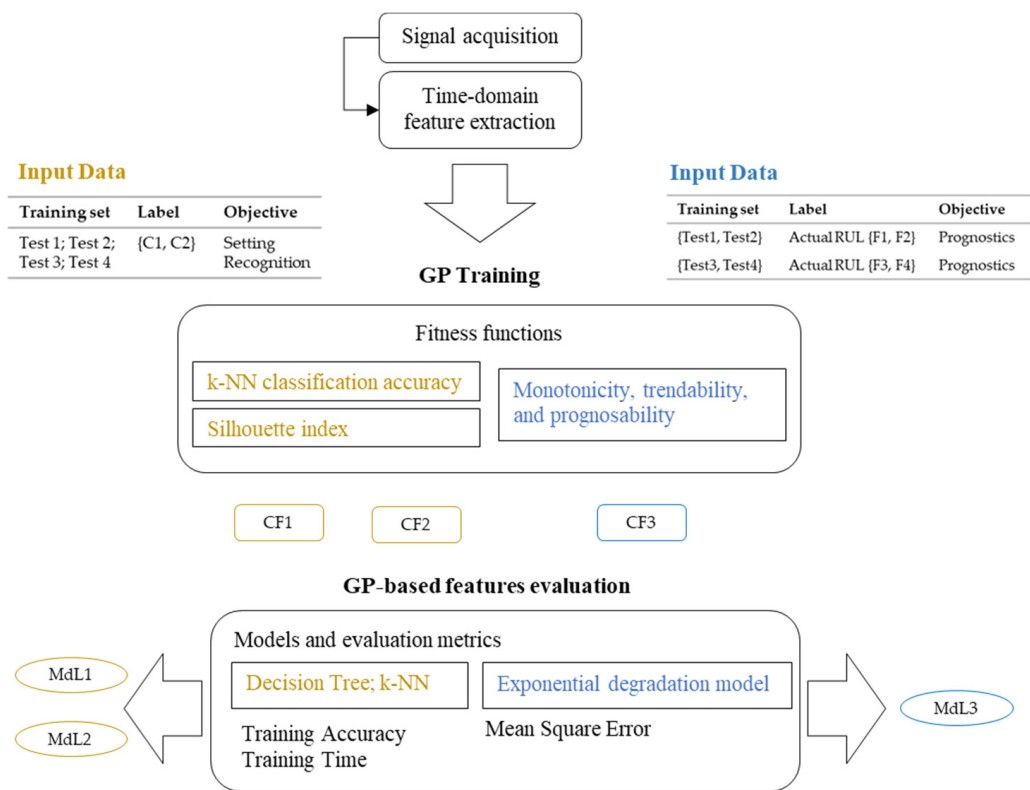

**Figure 2.** Framework for GP training.

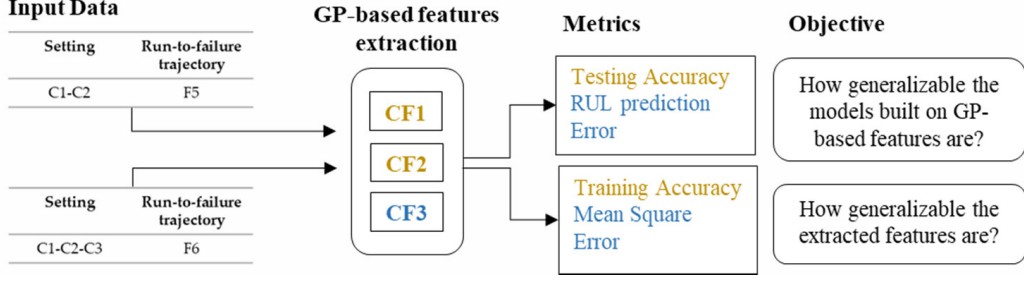

**Figure 3.** Framework for GP-based feature evaluation (testing).

After signal acquisition under conditions described in Table 4, nine typical time-domain features, summarized in Table 5, are extracted for each signal of tests 1, 2, 3, and 4 used for the GP training step. As the system is provided with three triaxial accelerometers, the original feature set contains 81 features. The length of the signal segments for feature

extraction is equal to 1 s (12,800 observations). Two different datasets are created from the so-obtained feature set and used for the setting recognition and the belt prognostics. Both datasets contain all the extracted features plus the label for each observation. The label may assume only two values corresponding to setting C1 and C2 for the setting recognition goal. The label C1 is assigned to tests 1 and 2, while the label C2 is assigned to tests 3 and 4. The label assumes continuous values for prognostics and expresses the second in which the observation has been collected, starting from 0. As feature observations are extracted every 1 s, the label increases by one along with the dataset, and the last value corresponds to the belt's duration. Note that the two training sets are used separately for the GP training to find two distinct HIs for the two machinery settings.

**Table 5.** Extracted time features from raw signals. $x_i$, is the signal (observation) at time $i$, $i = 1, \ldots, S_N$, $S_N$ is the length of the signal segments, $\overline{x}$ is the mean value of $x_i$, $i = 1 \ldots S_N$, and $\sigma^3$ $\sigma^4$ are third and the fourth moment of $x_i$, $i = 1 \ldots S_N$, respectively.

| Feature Name | Feature Formula |
|:---:|:---:|
| Peak | $f_{Peak} = x_{max} = max|x_i|$ |
| Peak-to-peak | $f_{Peak2Peak} = |max(x_i) - min(x_i)|$ |
| Mean | $f_{Mean} = \frac{1}{N} \sum_{i=1}^{N} x_i$ |
| Root mean square (RMS) | $f_{RMS} = \sqrt{\frac{1}{N} \sum_{i=1}^{N} x_i^2}$ |
| Crest factor (CF) | $f_{CrestF} = f_{Peak} / f_{RMS}$ |
| Kurtosis | $f_{Kurt} = \frac{1}{N} \sum_{i=1}^{N} \frac{(x_i - \overline{x})^4}{\sigma^4}$ |
| Skewness | $f_{Skew} = \frac{1}{N} \sum_{i=1}^{N} \frac{(x_i - \overline{x})^3}{\sigma^3}$ |
| Shape factor | $f_{ShapeF} = f_{RMS} / f_{Mean}$ |
| Impulse factor | $f_{ImpulseF} = f_{Peak} / f_{Mean}$ |

The dataset created for system setting recognition is provided to train GP with a classification-based fitness function and a clustering-based fitness function. In the first case, a supervised approach is adopted during the training, while an unsupervised learning approach is adopted in the clustering-based GP. The datasets created for prognostics are provided to a GP aiming to construct a component-level feature revealing the degradation processes of the belt during the run-to-failure trajectories F1, F2, F3, and F4. The output of the training phase is represented by three features, CF1, CF2, and CF3, that correspond to the best feature for the classification-based GP, clustering-based GP, and GP for HI construction, respectively.

The performances of the features extracted for setting recognition, i.e., CF1 and CF2, are evaluated in terms of training accuracy and training time provided through two typical classification models, i.e., the DT and the k-NN [36]. The performance of the constructed feature for prognostics, i.e., CF3, is evaluated by considering its ability to predict the belt Remaining Useful Life. First, the obtained HI for each system setting is smoothed through a moving average method. Then, an exponential model is applied to one of the available trajectories for each condition, and the accuracy of RUL prediction is computed through Equation (2), which represents the mean prediction error.

$$meanerror\% = \frac{1}{N} \sum_{n=1}^{N} \frac{|realRUL_n - predictedRUL_n|}{realRUL_n} \times 100 \qquad (2)$$

where $N$ is the number of observations.

The testing phase has two main goals, i.e., to assess the generalization ability of the models built on the constructed features and to assess the ability of constructed features to recognize novel settings [37], where novel means that they are not included in the GP training process. Therefore, all the CFs are extracted from two other datasets during the testing phase. The first dataset includes the trajectory F5, collected under known system settings (C1 and C2), while the second dataset includes trajectory F6, collected under

known settings C1 and C2, and novel setting C3. The generalization ability of the models is assessed using the classification models built during the training phase to make predictions on test 5 and compute the prediction accuracy (accuracy for setting recognition and RUL prediction error for prognostics). The generalization ability of constructed features is assessed by training classification and exponential degradation models on test 6, which includes three settings and evaluating the training accuracy and the Mean Square Error.

Note that the k-NN in the fitness function of the classification-based GP is trained using the cross-validation method. Standard values of 80% of observations for the training set and 20% for the testing set have been used. Instead, the k-means algorithm performed in the clustering-based GP is iterated five times for each GP iteration to get the best data partitioning. In addition, the DT and k-NN used to evaluate the different feature sets are always trained using a 5-fold cross-validation. Finally, it is worth pointing out that the testing accuracy computed during the testing phase of the methodology refers to the prediction accuracy of models built in the training phase and not to the testing accuracy resulting from the cross-validation.

The fitness functions used for training GPs are described in the following subsection.

*2.3. GP Fitness Functions*

The GP-OLS MATLAB toolbox [38], with a modified fitness function, has been used for GPs training. In particular, three fitness functions are considered. In the case of the supervised approach, the accuracy of k-NN is considered [31]. In the case of the unsupervised approach, the fitness function is equal to the mean value of the silhouette coefficient computed on the data partitions provided by the k-means algorithm. The silhouette coefficient, defined in [29] and given in Equation (3), measures the similarity of a point to the points belonging to the same cluster compared with points belonging to the other clusters.

$$Fit = \frac{1}{N} \sum_{i=1}^{N} s_i \tag{3}$$

where

1.  $s_i = \frac{a_i - b_i}{\max\{a_i, b_i\}}$ is the silhouette value of the point $i$
2.  $a_i = \frac{1}{|C_i| - 1} \sum_{j \in C_i, i \neq j} d_{i,j}$ is the average distance from the ith point to the other points in the same cluster as $i$
3.  $b_i = \min_{k \neq i} \frac{1}{|C_k|}$ is the minimum average distance from the ith point to points belonging to other clusters
4.  $N$ is the total number of observations

The silhouette value ranges from $-1$ to 1. A high silhouette value indicates that $i$ is well matched to its cluster and poorly matched to other clusters. The clustering solution is appropriate if most points have a high silhouette value.

The fitness function used to construct an optimal HI, as shown in Equation (4), consists of three terms: monotonicity, trendability, and prognosability.

$$Fit = w_1 Mon + w_2 Trend + w_3 Progn \tag{4}$$

where *Mon*, *Trend*, *Progn* are calculated by Equations (5)–(7), respectively [39]; $w_1$, $w_2$, $w_3$ are the weights associated with each HI evaluation criteria, and $\sum_i w_i = 1$.

$$Mon = \frac{1}{N_F} \left| \sum_{i=1}^{N} \left( \frac{n_i^+}{n_i - 1} - \frac{n_i^+}{n_i - 1} \right) \right| \tag{5}$$

$$Trend = min(|corrcoeff_{ij}|), \; i,j = 1, \dots, N_F \tag{6}$$

$$Progn = \exp\left(-\frac{std\left(HI_{fail}\right)}{mean\left(\left|HI_{start} - HI_{fail}\right|\right)}\right) \tag{7}$$

where $N_F$ is the number of run-to-failure trajectories, $n_i^+(n_i^-)$ indicates the number of observations characterized by a first positive (negative) derivative, $corrcoeff_{ij}$ is the linear correlation coefficient between the $i$th and $j$th run-to-failure trajectory.

The other parameters adopted for all GPs are summarized in Table 6.

**Table 6.** GP parameters set.

| Parameter | Value |
|---|---|
| Function set | Add, Subtraction, Protected division, Multiplication, Logarithm, Power, Exponential |
| Terminal set | Peak, Peak-to-peak, Mean, RMS, Crest factor, Kurtosis, Skewness, Shape factor, Impulse Factor |
| Population size | 100 |
| Max tree depth | 10 |
| Generation gap | 0.9 |
| $P_c$ | 0.9 |
| $P_m$ | 0.1 |
| Parents selection method | Tournament selection |
| Type of crossover | One point crossover |
| Replacement | Elitism |
| Number of generations | 30 |
| Termination criteria | Max. number of generations (iterations) |

## 3. Results

### 3.1. GP for System Setting Recognition

This section reports the best results obtained from 20 runs of each GP. In Table 7, the mean fitness value, the best fitness value, and the feature constructed at each iteration of the best run of classification-based and clustering-based GP are summarized.

**Table 7.** Fitness values and mathematical formulation of the feature constructed at each iteration of the best run of classification-based and clustering-based GP.

| Iter. | Classification | | | Clustering | | |
|---|---|---|---|---|---|---|
| | Mean Fitness Value | Best Fitness Value | FC1 | Mean Fitness Value | Best Fitness Value | FC2 |
| 1 | - | 0.950801 | $f_{Mean}(x.a1) \times f_{Skew}(x.a1) \times f_{Peak2Peak}(x.a3)$ | - | 0.720651 | $f_{Skew}(z.a2)$ |
| 2 | 0.840463 | 0.977063 | $f_{CrestF}(z.a1) + f_{Peak}(x.a1)$ | 0.711219 | 0.720651 | $f_{Skew}(z.a2)$ |
| 3 | 0.819037 | 0.977063 | $f_{CrestF}(z.a1) + f_{Peak}(x.a1)$ | 0.711219 | 0.720651 | $f_{Skew}(z.a2)$ |
| 4 | 0.841445 | 0.977517 | $f_{CrestF}(z.a1) + f_{Peak}(x.a1)$ | 0.741445 | 0.757926 | $f_{Skew}(z.a2) \times f_{RMS}(y.a3)$ |
| 5 | 0.662749 | 0.977517 | $f_{CrestF}(z.a1) + f_{Peak}(x.a1)$ | 0.753748 | 0.777517 | $f_{Peak2Peak}(x.a3) \times f_{Skew}(y.a2)$ |
| 6 | 0.693168 | 0.978424 | $f_{CrestF}(z.a1) + f_{Skew}(y.a2)$ | 0.753748 | 0.785372 | $f_{Peak2Peak}(x.a3) \times f_{Skew}(y.a2)$ |
| 7 | 0.931907 | 0.998413 | $f_{Skew}(y.a2) + f_{Peak}(x.a1)$ | 0.753748 | 0.795372 | $f_{RMS}(y.a3) \times f_{Skew}(y.a2)$ |
| 8 | 0.947476 | 0.998526 | $f_{Skew}(y.a2) + f_{Peak}(x.a1)$ | 0.767290 | 0.815372 | $f_{RMS}(y.a3) \times f_{Skew}(y.a2)$ |
| 9 | 0.976685 | 0.998526 | $f_{Skew}(y.a2) + f_{Peak}(x.a1)$ | 0.734272 | 0.815372 | $f_{RMS}(y.a3) \times f_{Skew}(y.a2)$ |
| 10 | 0.976421 | 0.998526 | $f_{Skew}(y.a2) + f_{Peak}(x.a1)$ | 0.746396 | 0.823728 | $f_{RMS}(y.a3) \times f_{Skew}(y.a2) + f_{Mean}(y.a2)$ |
| 11 | 0.998602 | 0.999698 | $f_{Skew}(y.a2) + f_{Kurt}(z.a1)$ | 0.745824 | 0.823728 | $f_{RMS}(y.a3) \times f_{Skew}(y.a2) + f_{Mean}(y.a2)$ |
| 12 | 0.998451 | 0.999698 | $f_{Skew}(y.a2) + f_{Kurt}(z.a1)$ | 0.797367 | 0.823728 | $f_{RMS}(y.a3) \times f_{Skew}(y.a2) + f_{Mean}(y.a2)$ |
| 13 | 0.956545 | 0.999811 | $f_{Skew}(y.a2) + f_{CrestF}(x.a2)$ | 0.796293 | 0.823728 | $f_{RMS}(y.a3) \times f_{Skew}(y.a2) + f_{Mean}(y.a2)$ |
| 14 | 0.952766 | 0.999811 | $f_{Skew}(y.a2) + f_{CrestF}(x.a2)$ | 0.817364 | 0.839811 | $f_{RMS}(y.a3) \times f_{Skew}(y.a2) + f_{Mean}(z.a2)$ |
| 15 | 0.956734 | 0.999811 | $f_{Skew}(y.a2) + f_{CrestF}(x.a2)$ | 0.826378 | 0.840800 | $f_{RMS}(y.a3) \times f_{Skew}(y.a2) + f_{Mean}(z.a2)$ |
| 16 | 0.999773 | 0.999849 | $f_{Skew}(y.a2) + f_{RMS}(x.a2)$ | 0.826738 | 0.840800 | $f_{RMS}(y.a3) \times f_{Skew}(y.a2) + f_{Mean}(z.a2)$ |
| 17 | 0.970602 | 0.999849 | $f_{Skew}(y.a2) + f_{RMS}(x.a2)$ | 0.826647 | 0.840800 | $f_{RMS}(y.a3) \times f_{Skew}(y.a2) + f_{Mean}(z.a2)$ |

**Table 7.** *Cont.*

| Iter. | Classification | | | Clustering | | |
|---|---|---|---|---|---|---|
| 18 | 0.999849 | 0.999887 | $f_{Skew}(y.a2) + f_{RMS}(x.a1)$ | 0.826849 | 0.840800 | $f_{RMS}(y.a3) \times f_{Skew}(y.a2) + f_{Mean}(z.a2)$ |
| 19 | 0.999735 | 0.999887 | $f_{Skew}(y.a2) + f_{RMS}(x.a1)$ | 0.826354 | 0.840800 | $f_{RMS}(y.a3) \times f_{Skew}(y.a2) + f_{Mean}(z.a2)$ |
| 20 | 0.999811 | 0.999924 | $f_{Skew}(y.a2) + f_{RMS}(x.a1)$ | 0.826354 | 0.840800 | $f_{RMS}(y.a3) \times f_{Skew}(y.a2) + f_{Mean}(z.a2)$ |
| 21 | 0.958472 | 0.999924 | $f_{Skew}(y.a2) + f_{RMS}(x.a1)$ | 0.826937 | 0.840800 | $f_{RMS}(y.a3) \times f_{Skew}(y.a2) + f_{Mean}(z.a2)$ |
| 22 | 0.999849 | 0.999924 | $f_{Skew}(y.a2) + f_{RMS}(x.a1)$ | 0.826929 | 0.840800 | $f_{RMS}(y.a3) \times f_{Skew}(y.a2) + f_{Mean}(z.a2)$ |
| 23 | 0.999849 | 0.999924 | $f_{Skew}(y.a2) + f_{RMS}(x.a1)$ | 0.825289 | 0.840800 | $f_{RMS}(y.a3) \times f_{Skew}(y.a2) + f_{Mean}(z.a2)$ |
| 24 | 0.999849 | 0.999924 | $f_{Skew}(y.a2) + f_{RMS}(x.a1)$ | 0.817839 | 0.840800 | $f_{RMS}(y.a3) \times f_{Skew}(y.a2) + f_{Mean}(z.a2)$ |
| 25 | 0.661087 | 0.999924 | $f_{Skew}(y.a2) + f_{RMS}(x.a1)$ | 0.826273 | 0.840800 | $f_{RMS}(y.a3) \times f_{Skew}(y.a2) + f_{Mean}(z.a2)$ |
| 26 | 0.999887 | 0.999924 | $f_{Skew}(y.a2) + f_{RMS}(x.a1)$ | 0.826142 | 0.840800 | $f_{RMS}(y.a3) \times f_{Skew}(y.a2) + f_{Mean}(z.a2)$ |
| 27 | 0.999735 | 0.999924 | $f_{Skew}(y.a2) + f_{RMS}(x.a1)$ | 0.826039 | 0.840800 | $f_{RMS}(y.a3) \times f_{Skew}(y.a2) + f_{Mean}(z.a2)$ |
| 28 | 0.999887 | 0.999924 | $f_{Skew}(y.a2) + f_{RMS}(x.a1)$ | 0.826377 | 0.840800 | $f_{RMS}(y.a3) \times f_{Skew}(y.a2) + f_{Mean}(z.a2)$ |
| 29 | 0.999849 | 0.999924 | $f_{Skew}(y.a2) + f_{RMS}(x.a1)$ | 0.826371 | 0.840800 | $f_{RMS}(y.a3) \times f_{Skew}(y.a2) + f_{Mean}(z.a2)$ |
| 30 | 0.961797 | 0.999924 | $f_{Skew}(y.a2) + f_{RMS}(x.a1)$ | 0.826352 | 0.840800 | $f_{RMS}(y.a3) \times f_{Skew}(y.a2) + f_{Mean}(z.a2)$ |

Figure 4 shows the final tree corresponding to the best solution and the fitness values at each iteration of the GP in the case of classification-based GP.

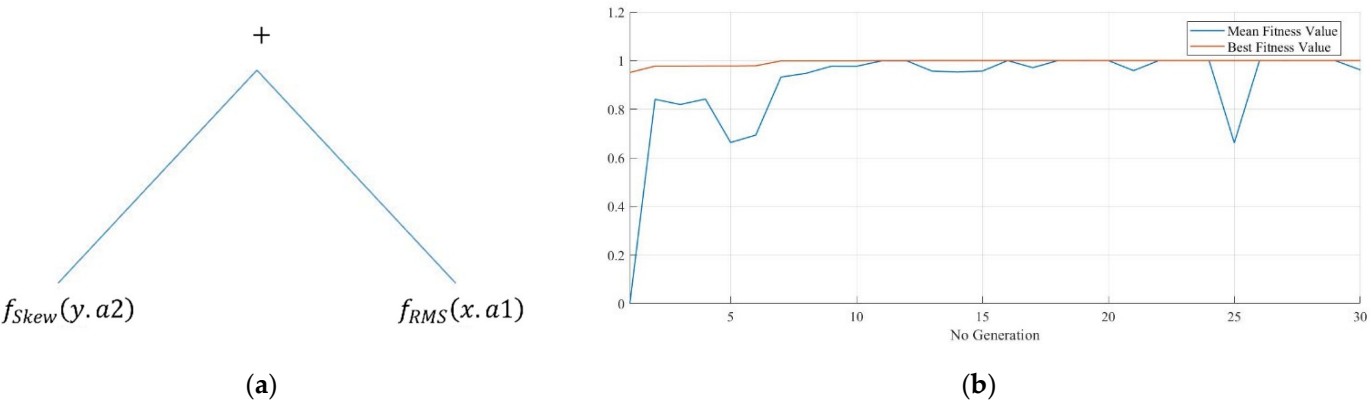

(**a**)                                    (**b**)

**Figure 4.** Classification-based GP results: (**a**) The tree representing the constructed features, and (**b**) the mean and the best fitness values at each GP iteration.

For classification, the best fitness value equal to 0.9999 is provided by the feature given in Equation (8) and shown in Figure 5a.

$$f_{Skew}(y.a2) + f_{RMS}(x.a1) \tag{8}$$

where, $f_{Skew}(y.a2)$ corresponds to the skewness extracted from the fifth signal (the *y*-axis of the second accelerometer) and $f_{RMS}(x.a1)$ corresponds to the RMS of the *x*-axis of the first accelerometer.

For clustering, the best fitness value equal to 0.8408 is provided by the feature in Equation (9) and shown in Figure 5b.

$$f_{RMS}(y.a3) \times f_{Skew}(y.a2) \times f_{Mean}(z.a2) \tag{9}$$

where $f_{RMS}(y.a3)$ is the RMS of the *x*-axes of the third accelerometer, $f_{Skew}(y.a2)$ is the skewness extracted from the *y*-axis of the second accelerometer, and $f_{Mean}(z.a2)$ is the mean extracted from the z-axes of the second accelerometer.

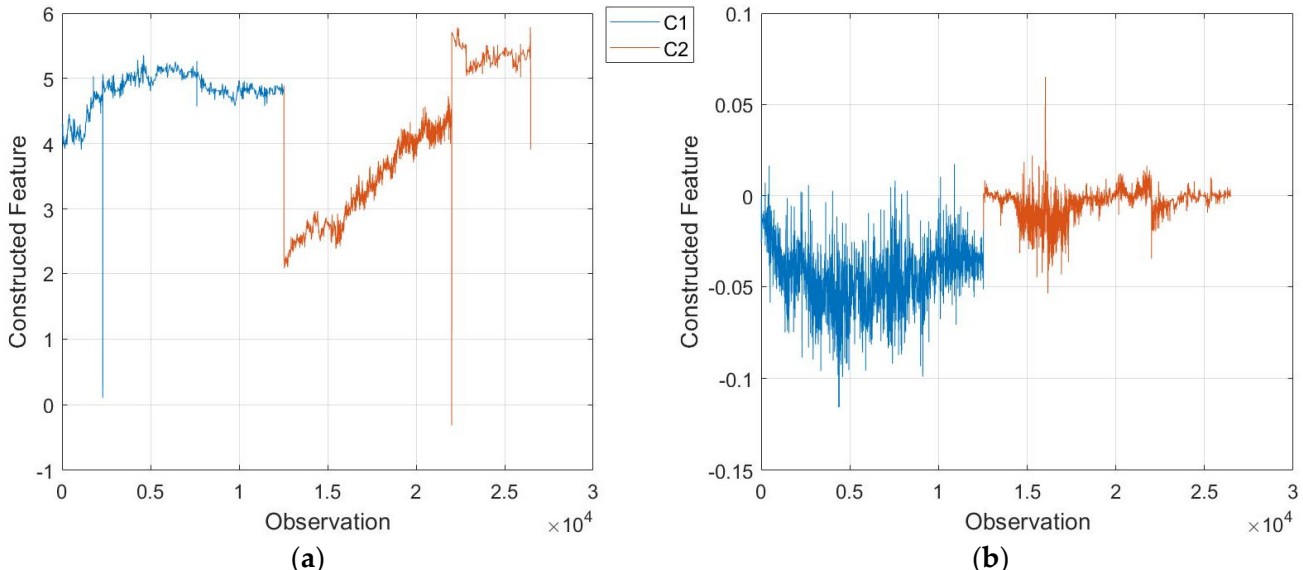

**Figure 5.** Constructed features through (**a**) classification-based GP and (**b**) clustering-based GP. The black line refers to the first setting trajectories (F1–F2). The red line refers to the second setting trajectories (F3–F4).

The DT and the k-NN are trained to evaluate the performance of the constructed features. Then, the obtained model is applied to the trajectory F5 and prediction accuracy is computed.

Both classification-based and clustering-based GPs provide high accuracy prediction during the training phase, while the accuracy is relatively low when applying the models to the testing trajectory.

DT and k-NN are also applied to the whole set of extracted features to evaluate the effect of feature construction on classification accuracy. Finally, GP-based feature construction is also compared with the Principal Component Analysis (PCA) (feature learning) [40] and the ReliefF (feature selection) [41]. For comparison, one only feature is selected for both PCA and ReliefF. Table 8 summarizes the obtained results, where the training accuracy is computed on trajectories F1–F4 and the testing accuracy on trajectory F5. It can be seen that the training accuracy is lower than that provided by GP-based features, meaning that both the Principal Component (PC) constructed through the PCA and the feature selected through the ReliefF algorithm are not able to separate the two classes well. In addition, the model accuracy prediction on test 5 is lower than that obtained by applying the models trained with GP-based features. Similarly, although considering all features as input of classification models provides the best training accuracy, the testing accuracy is low, and the training time is higher, especially with k-NN.

**Table 8.** Classification performance.

| Method for Feature Extraction | Model | Training Accuracy (%) | Training Time (s) | Testing Accuracy (%) |
|---|---|---|---|---|
| Classification-based GP | DT | 90.4 | 2.6 | 67.03 |
| | k-NN | 90.8 | 1.25 | 67.03 |
| Clustering-based GP | DT | 96.0 | 0.71 | 67.30 |
| | k-NN | 94.5 | 0.72 | 66.85 |
| No feature selection | DT | 99.99 | 3.53 | 31.52 |
| | k-NN | 100 | 56.94 | 61.14 |
| PCA | DT | 81.9 | 1.78 | 61.30 |
| | k-NN | 74.9 | 1.63 | 57.44 |
| ReliefF | DT | 74.8 | 1.93 | 61.79 |
| | k-NN | 64.1 | 1.57 | 52.86 |

Finally, the performance of the extracted features for classification and clustering are also evaluated based on the ability to distinguish unknown conditions. The time-domain features extracted from raw signals collected during test 6, which considers the system under a "novel" operating condition (F6), are added to the training set in the feature evaluation step of the methodology. In Figure 6, the constructed features with the clustering GP-based (left) and the classification GP-based (right) are shown, where distinct colors represent distinct settings. The results of classification models trained on these datasets are summarized in Table 9, including the training accuracy obtained using the features obtained through the PCA and the ReliefF.

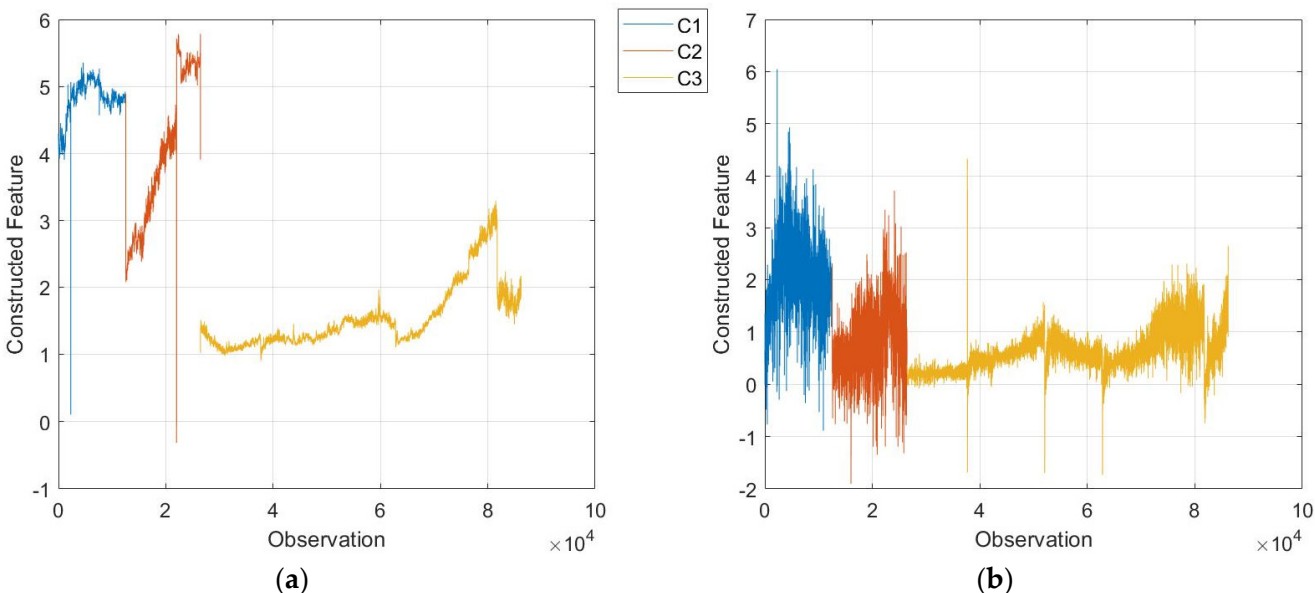

**Figure 6.** Constructed features through (**a**) clustering-based GP and (**b**) classification-based GP with an additional operating condition.

**Table 9.** Classification accuracy (%) obtained with GP-based FCs, and feature extracted through PCA and ReliefF after adding a third machinery setting into the training set.

| Model | Classification-Based GP | Clustering-Based GP | PCA | ReliefF |
|---|---|---|---|---|
| DT | 92.8% | 81.1% | 92.2% | 76.4% |
| k-NN | 90.8% | 70.8% | 89.3% | 69.1% |

From Table 9, it can be seen that when the system-level feature is constructed with the classification-based GP, classification models trained with the additional operating condition provide a class prediction accuracy higher than 90%. On the contrary, the classification accuracy is lower than 90% when using the clustering-based GP.

In addition, ReliefF provides lower training accuracy than GP-based features, while the PCA produces similar results as the classification-based GP. The direct conclusion of this result is that feature construction provides more discriminant features than feature selection.

### 3.2. GP for Belt Prognostics

After system-level feature construction, a component-level HI (for the belt wear) has been constructed through the GP with the fitness function expressed by Equation (4). As the acquired signals have acceptable values of prognosability and trendability, greater weight is assigned to the monotonicity in order to obtain more monotonic HIs. Therefore, the weight of the monotonicity, $w_1$, is set to 0.5 while both other weights are set to 0.25.

As in the first case, the training set is made of run-to-failure trajectories F1, F2, F3, and F4. However, these trajectories have been divided into two input sets to extract an HI

for each condition. The fitness values of constructed HIs for both conditions during some GP runs are summarized in Table 10, including the RUL prediction mean error computed through Equation (2) on the signal acquired during test 5. As test 5 includes both C1 and C2 conditions, the error has been computed using $HI_{C1_i}$, $i = 1, \ldots, 4$, until the first setting is implemented and $HI_{C2_i}$, $i = 1, \ldots, 4$ for the remaining observations.

**Table 10.** HI construction GP.

| Constructed HI | Fitting Function | Mean Error (%) |
|---|---|---|
| $HI_{C1_1} = f_{Skew}(x.a1)^{(f_{Kurt}(z.a1) - f_{RMS}(y.a1))}$ | 0.89 | 14.2 |
| $HI_{C1_2} = f_{Peak}(x.a3) / f_{Mean}(y.a1)$ | 0.87 | 33.4 |
| $HI_{C1_3} = f_{CrestF}(y.a3) / f_{RMS}(y.a2)$ | 0.78 | 40.5 |
| $HI_{C1_4} = f_{ImpulseF}(z.a2) / e^{f_{Peak}(z.a3)}$ | 0.94 | 130.1 |
| $HI_{C2_1} = f_{Peat2Peak}(x.a2) / e^{f_{RMS}(x.a1)}$ | 0.985 | 36.6 |
| $HI_{C2_2} = f_{Skew}(x.a3) \times f_{RMS}(z.a3) \times \log\left(\frac{f_{ImpulseF}(y.a3)}{\log f_{Mean}(y.a3)}\right)$ | 0.87 | 10.5 |
| $HI_{C2_3} = f_{RMS}(z.a3) - \log\left(\frac{f_{RMS}(x.a3)}{f_{ShapeF}(z.a3)}\right)$ | 0.99 | 30.5 |
| $HI_{C2_4} = f_{Skew}(x.a3)$ | 0.78 | 35 |

Each constructed HI represents an optimal solution according to the GP fitness function. However, when a degradation model is built to compute the RUL, not all HIs provide good RUL prediction mean error results. For instance, $HI_{C1_4}$ and $HI_{C2_3}$ have the best fitness values, equal to 0.94 and 0.99, but they provide a high RUL mean prediction error. On the contrary, $HI_{C1_1}$ and $HI_{C2_2}$ provide a slightly lower fitness value, but a lower RUL mean prediction error. Hence, $HI_{C1_1}$ and $HI_{C2_2}$ are chosen as the best HIs for setting 1 and setting 2, respectively, since they provide the best fitness value among HIs with an acceptable prediction error. In this case, an error between 10% and 20% is acceptable because computed from the beginning of the test, when the belt was still in good health conditions. The constructed HIs for each trajectory of the training dataset are depicted in Figure 7. Table 11 summarizes the values of monotonicity, trendability, and prognosability of the two constructed HIs.

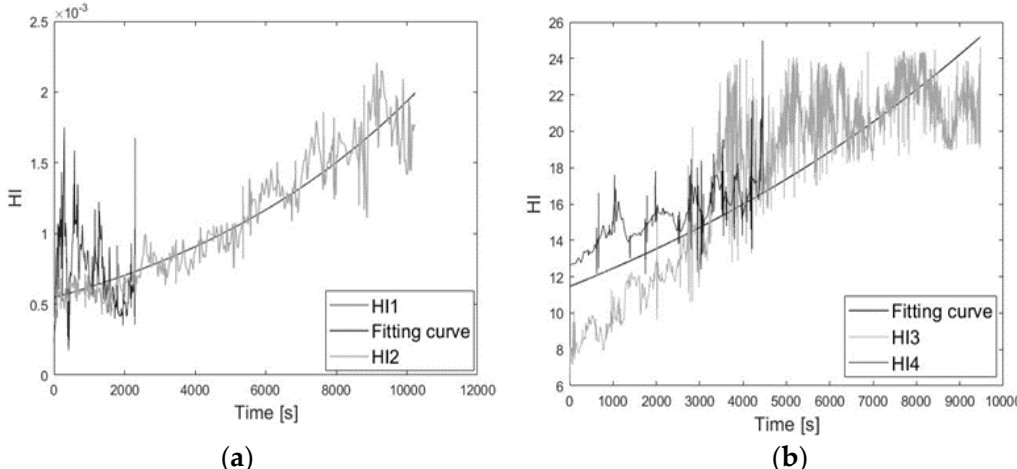

**Figure 7.** Constructed HIs for (**a**) setting 1 and (**b**) setting 2.

**Table 11.** Monotonicity, trendability, and prognosability values of His constructed through GO for the two machinery settings.

| Health Indicator | Monotonicity | Trendability | Prognosability |
|---|---|---|---|
| $HI_{C1_1}$ | 0.3999 | 0.1042 | 0.9095 |
| $HI_{C2_2}$ | 0.2637 | 0.7599 | 0.7599 |

To demonstrate the advantage of constructing HIs through GP, they are compared to a typical HI used for prognostics: the correlation coefficient between nominal signals and fault signals [42]. Table 12 summarizes the length of nominal signals and fault signals considered for each run-to-failure trajectory.

**Table 12.** Length of nominal and fault signals considered for the correlation coefficient computation.

| Run-to-Failure Trajectory | Nominal (s) | Fault (s) |
|:---:|:---:|:---:|
| F1 | 763 | 763 |
| F2 | 1.389 | 6.945 |
| F3 | 1.531 | 6.124 |
| F4 | 1.088 | 4.352 |

The correlation coefficient has been computed for each acquired signal, and their monotonicity, trendability, and prognosability values are summarized in Table 13. As can be seen, all metrics have lower values than those computed on constructed HIs with GP.

**Table 13.** Length of nominal and fault signals considered for the correlation coefficient computation.

| Setting | Metric | X-a1 | Y-a1 | Z-a1 | X-a2 | Y-a2 | Z-a2 | X-a3 | Y-a3 | Z-a3 |
|:---:|:---|:---:|:---:|:---:|:---:|:---:|:---:|:---:|:---:|:---:|
| | Monotonicity | 0.0045 | 0.0154 | 0.0036 | 0.0041 | 0.0035 | 0.0080 | 0.0138 | 0.0147 | 0.0160 |
| C1 | Trendability | 0.0040 | 0.0077 | 0.0011 | 0.0311 | 0.0241 | 0.0262 | 0.0074 | 0.0083 | 0.0173 |
| | Prognosability | 0.1997 | 0.8400 | 0.8152 | 0.8247 | 0.9104 | 0.3104 | 0.6745 | 0.8328 | 0.4767 |
| | Monotonicity | 0.0133 | 0.0060 | 0.0301 | 0.0093 | 0.0031 | 0.0071 | 0.0041 | 0.0072 | 0.0033 |
| C2 | Trendability | 0.0503 | 0.0222 | 0.1513 | 0.0090 | 0.0414 | 0.0589 | 0.0242 | 0.0313 | 0.0150 |
| | Prognosability | 0.2828 | 0.3350 | 0.2577 | 0.2569 | 0.0632 | 0.3011 | 0.2316 | 0.8701 | 0.3361 |

### 3.3. Discussion

This section describes the three primary outcomes of the present work.

First, as shown in Table 8, the training accuracy obtained with GP-based feature construction is higher than the training accuracy obtained considering the feature selected through the ReliefF and the feature extracted through the PCA. This result implies that the constructed features FC1 and FC2 well separate the two classes corresponding to the settings. On the contrary, using all the time-domain features provides a better training accuracy. However, the so-built models overfit the training data. Indeed, the testing accuracy is equal to 31% in the case of DT and 61% in the case of k-NN. Similar testing accuracies are obtained with PCA and ReliefF as well. On the contrary, the testing accuracies obtained by using the GP features are higher than 67%. Therefore, the features FC1 and FC2 separate the two settings and allow the building of a more generalizable classification model.

Second, as shown in Table 9, when GP features are used to train classification models with three classes, classification accuracy is higher than using the extracted feature through the ReliefF. This result implies that FC is better than feature selection for novelty detection. Indeed, only selected features are used as input trained models in streaming diagnostics and prognostics. This means that if a feature does not reveal an unknown setting, the change cannot be detected.

Finally, concerning the GP trained to extract a HI for the belt prognostics, results show that the constructed HIs have similar trends for failure occurred in the same operating condition. However, the trend is not strictly monotonic, which makes it hard to build a robust degradation model for RUL prediction. Indeed, no HI produces a mean RUL prediction error lower than 10%. However, the constructed HI is compared with the correlation coefficient between a nominal signal and the two fault signals collected during two different settings. This comparison shows that correlation coefficients have lower values of monotonicity, prognosability, and trendability than the constructed HI through the GP. Therefore, the constructed HIs are more suitable for component-level prognostics.

## 4. Conclusions

This paper applies Genetic Programming to construct system-level and component-level features for system setting recognition and prognostics. First, the use of Genetic Programming in the PHM field is reviewed, and the essential elements of GP are described. Then, three GPs with distinct fitness functions are trained using vibration signals collected from an experimental platform built in the Department of Industrial Engineering laboratory of the University of Bologna.

The first two GPs aim to construct a feature that reveals the system setting implemented during signal acquisition. The first GP adopts a supervised ML algorithm in the fitness function, while the second adopts an unsupervised learning approach. The two approaches can be applied depending on the available training data. The supervised approach may be adopted if the training data are provided with a label indicating the system setting. Otherwise, the unsupervised approach is necessary. This latter situation is widespread among automatic machinery producers, who do not know all machinery use conditions and collect the data from their clients, who, in turn, often provide unlabelled datasets. Two traditional ML models, i.e., the Decision Tree and the k-Nearest Neighbor, are chosen to evaluate the performance of the constructed features in terms of training accuracy, training time, and model accuracy when used to predict the setting of observations collected after the training, named testing accuracy. Finally, the GP features are also evaluated against the ability to reveal a setting not included during the feature construction process. The third GP aims to construct an optimal HI for the belt degradation process. The fitness function is computed as a weighted mean of monotonicity, trendability, and prognosability.

Future development of this work will include a multi-objective GP for HI construction, which also considers HI evaluation criteria directly correlated with the effectiveness of the prognostic results. Moreover, an anomaly detection algorithm for incipient degradation detection will be applied to define the instant in which the algorithm should begin the RUL computation in a real-time application.

**Author Contributions:** Conceptualization, F.C. and R.P.; methodology, F.C. and F.G.G.; software, R.P.; validation, A.R. and M.B.; writing—original draft preparation, F.C.; writing—review and editing, M.B. and A.R.; supervision. F.G.G. All authors have read and agreed to the published version of the manuscript.

**Funding:** This research received no external funding.

**Informed Consent Statement:** Not applicable.

**Data Availability Statement:** The data presented in this study are available on request from the corresponding author. The data are not publicly available due to agreements with the department.

**Conflicts of Interest:** The authors declare no conflict of interest.

## Appendix A

Table A1 summarizes the technical aspects of all elements included in the test rig.

**Table A1.** Technical characteristics of components included in the test rig.

| Component | Characteristics | Values | Component | Characteristics | Values |
|---|---|---|---|---|---|
| | Number of teeth | 30 | | Number of teeth | 60 |
| | Pitch | 5 mm | | Module | 1 |
| Pulley 1 | To suit belt width | 10 mm | Spur gear 1 | Pitch | 60 mm |
| | Bore | 8 mm | | Bore | 10 mm |
| | Material | Aluminum | | Material | Steel |

**Table A1.** *Cont.*

| Component | Characteristics | Values | Component | Characteristics | Values |
|---|---|---|---|---|---|
| Pulley 2 | Number of teeth<br>Pitch<br>To suit belt width<br>Bore<br>Material | 40<br>5 mm<br>10 mm<br>8 mm<br>Aluminum | Spur gear 2 | Number of teeth<br>Module<br>Pitch<br>Bore<br>Material | 120<br>1<br>120 mm<br>12 mm<br>Steel |
| Belt | Number of teeth<br>Pitch<br>Length<br>Width<br>Maximum speed<br>Material | 122<br>1.2<br>5 mm<br>610 mm<br>10 mm<br>80 m/s<br>Polyurethan | Ball-bearing | Inside diameter<br>Outside diameter<br>Static load rating<br>Material | 20 mm<br>47 mm<br>6.55 kN<br>Steel |
| Long Closed Bush Shaft | Length<br>Diameter<br>Hardness<br>Tolerance<br>Material | 1 m<br>20 mm<br>60→64 HRC<br>h6<br>Steel | | | |

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
