# Peer review of "Genetic Programming-Based Feature Construction for System Setting Recognition and Component-Level Prognostics"

_applsci, doi:10.3390/app12094749_

Round 1
Reviewer 1 Report
The paper addresses the problem of feature extraction in raw signals using Genetic Programming (GP) for Prognostics and Health Management (PHM). The authors construct classification and clustering models for a test rig based on GP. A component-level Health Indicator (HI) is built for the belt wear in the system considering: the features found by GP, monotonicity, trendability, and prognosability. Results show the potential of GP for Feature Construcion and the advantages of the clustering-based GP in recognizing novel operating conditions.
The topic of the paper is interesting. However, some issues are not clear, and they should be improved. In general, the abstract does not highlight the advantages of the proposed design, and the conclusions do not summarize the main benefits of the approach; some numerical values can solve both issues. The methodology contains several unclear elements in configuration, training, testing, and results.
The related works section should summarize (in a table) the main similarities and differences of the proposed approach concerning other methods. A diagram of the “test rig” can simplify its comprehension because Figure 1 (picture) contains many elements that are not part of the system. An appendix might define the technical aspects of the elements in the test ring. A more detailed explanation of the methodology (Figure 2) can enhance the understanding of the research.
For the training and testing model, the authors define 12,800 observations, but they do not establish the number (with and without failure) and types (F1, F2, etc.) of observations used in the training and testing process (datasets), for example, Cross-Validation is a standard methodology to do it. The description and difference between failures (from F1 to F6) are not clear (how does the system fail?). The authors should introduce references to these failures in the industry to validate the environment.
The authors should provide statistics about the fitness values for classification and clustering (average fitness value of the population during each iteration, initial and final populations, the best solutions for all experiments, etc.). Additional, some examples (figures) of the GP codification and the genetic operators. The notations for Eq. 8, Eq. 9, and Table 12 are confusing; they must use more descriptive and straightforward descriptions. For example, skewness(y, a2) + RMS(x, a1) instead of ?42+?4 for the skewness of the y-axis in the second accelerometer plus RMS of the x-axis of the first accelerometer.
Several aspects of the configuration setup should be justified (the authors propose these values, or they use typical values): AC motor speed and breaking force (Table 2); monotonicity, trendability, and prognosability (0.5, 0.25, and 0.25); GP parameters (population size, max tree depth, Pc, Pm, number of generations, and runs). The classification accuracy of PCA and ReliefF have to be compared in Table 11 to show the advantages of the proposed method. Exponential function (e) is not defined in the function set (Table 5), but it appears in ???14 and ???21 (Table 12).
There is no reference value to compare the efficiency of HIs generated by the GP in Table 12 (another method different from GP). Moreover, the dataset used to compute the “fitting value” and “mean error” is not described. The election of ???11 and ???22 looks a little arbitrary (opinion of the authors) because it is a multi-objective problem max(Fitting Value) and min (Mean error), the authors can justify their election of “the best trade-off” with a formal methodology or with a deeper explanation.
Proofreading is suggested since some grammar errors and typos need to be corrected. Some instances are:
- Authors: All the authors are from the same institution (delete the affiliation numbers), and add the word “and” between the last and one before the previous authors.
- Typos:
Line 119, “Finally, A”,
Line 162, “Moltiplication”,
Line 347, “4 PCs”
Line 376, “His”.
- Indentation: Lines 144 and 294.
- Numbers: Some float numbers use comma instead of point (0,5, 0,25, 0,9999, and 0,8408).
- Figures: low-quality images and vertical axis without title (Figures 3 and 4).
- References: Check the use of references from Line 176 to 202.
- Abbreviations: Use “k-NN” or “kNN” or “KNN” but not all of them, and “Principal Component Analysis (PCA)”.
- Author Contributions: The authors do not define their contribution to the document.
- Data Availability Statement: it is not defined.
- Tables: Table 4 does not define several symbols, Table 5 does not describe the type of mutation, and several tables contain similar information, and they can be joined (Tables 6 and 7, and Tables 8, 9, and 10).
- Equations: N describes the number of observations (Eq. 2) and the number of run-to-failure trajectories (Eq. 5 and 6).
In general, some sections of the document are difficult to follow.
Author Response
Thank you for your comments and valuable suggestions. Please, look at the attached file for responses.

Reviewer 2 Report
The authors proposed a new method for "Genetic Programming-based feature construction for system 2 setting recognition and component-level prognostics". In general the paper is well written, the methodology is well presented and the results of the experimental studies support the conclusions.
However in order to further enhance the quality of the paper, I have the following option suggestions.
- In section 1, introduction, the authors coould describe more about the relative merits of the proposed method as compared to the methods proposed in past research.
- In section Material and Method, a flow chart could be included to illustrate the logic of the procedures from line 137 to line 143.
- In section 2, the author could explain the reasons why the GP Genetic Programming is used in the proposed method (instead of other metahuristic (or other optimization) methods.
- A graph of fitness value vs. No. generation could be included in the section 3. results so as to illustrate how the fitness is improved when no. of generation increases.
Author Response

(The authors gave the same response as above.)

Reviewer 3 Report
The classification method used in this article is decision tree , KNN .The mentioned techniques are very old. There are a lot of advanced classification techniques are available. Why did the author choose these techniques ?Why not other techinques.
In section 2.1 the author mention "Figure 2. Methodology design ". In this model, author need to elaborate on every phase.
The author suggested comparing the existing work to the proposed design methodology. How has this article improved? Need more justification.
What is the significant of equations 5 to 7 ?
Author Response

(The authors gave the same response as above.)

Round 2
Reviewer 1 Report
The current version of the document covers several recommendations; however, there are some aspects than can be improved.
The authors do not justify the phrase “Results demonstrate a more generalization ability of GP-based features against other feature selection and learning methods”, they should provide a numerical value in other case this statement is only their opinion.
I think the discussion and the conclution should be described in different sections. Not all the information was updated according to the symbols in Table 5, for example: From Line 382 to Line 389, and elements in Table 10.
Proofreading is suggested since some grammar errors and typos need to be corrected. Some instances are:
- Name and affiliation: Authors should provide an abbreviation of their names and their emails, also the abbreviation should be used in “Author Contributions”
- Typos:
Line 258, “characteristics.For”
Line 374, “Figures 3 show s”
Line 284, “HIs” instead of “His”
Line 462, “figure 7”
- Numbers: several problems with numbers in Tables 4 and 7, and
Line 274, “12,800” instead of “12800”.
- Figures: Figure 4 is missing and the quality of some figures.
- Tables: It is not clear the values of FC1 and FC2 in Tables 7, for example: (x52), ((x23)+(x10)), also in figure 3(a).
- Abbreviations: Do not define several time the same acronymos, for example, Decision Tree (DT) and the k-Nearest Neighbor (k-NN) .
There are many errors in the added information, they should be solved before to accept the paper.
Author Response
Dear reviewer,
all authors thank you for your valuable suggestions. Please, find the response to your comments in the attached file.

Reviewer 3 Report
A lot of places grammatical mistakes are there
It seems some of the sections, sentences are not framed well .
Required to submit from native speaker English proofreading certificate
Author Response

(The authors gave the same response as above.)
